# The Application of Mott’s Distribution in the Fragmentation of Steel Coaxial Cylinders

**DOI:** 10.3390/ma16175783

**Published:** 2023-08-24

**Authors:** Octavian-Gabriel Chiriac, Florina Bucur, Adrian-Nicolae Rotariu, Eugen Trană

**Affiliations:** Military Technical Academy “Ferdinand I”, 050141 Bucharest, Romania; octachiriac@gmail.com (O.-G.C.); florina.bucur@mta.ro (F.B.); eugen.trana@mta.ro (E.T.)

**Keywords:** shell fragmentation, multilayered case, coaxial cylinders, Mott distribution, FEM simulation

## Abstract

This theoretical study analyzes the possibility to use the classical Mott’s hypothesis to model the natural fragmentation of cylindrical structures with two or more metal cylinders arranged coaxially. A critical analysis on the validity of the used hypothesis was conducted based on empirical relations and numerical simulations. The established algorithm allows the determination of a fragment mass scale parameter for each individual cylinder, which is why the cumulative distribution of fragments for the entire structure may be calculated. The results obtained for the structures with two and three cylinders, with equal masses or equal wall thicknesses, can be approximated using a modified Mott’s distribution formula in which the number of cylinders is used as an additional parameter.

## 1. Introduction

The large-scale use of modern high-explosive shells, starting with the First World War, represented the catalyst for the research dedicated to the interaction between shells bodies and the gases obtained by detonating the carried explosive charge. The interest is given by the fact that during this process, fragments are produced that move at very high speeds, which is why they represent a serious kinetic threat to the forces and the equipment of the opponent [1].

Regardless of the situation in which the detonation of the explosive charge occurs, due to the cylindrical shape, the metal body is propelled radially by the produced gases, being subjected to a rapid deformation process that leads to deformations beyond the material strength limits, a condition that allows fractures to occur. Experimental investigations have identified two types of fractures through which the projectile body fragmentation is achieved, some that start from the outer surface, determined by tensile stresses, and some that start from the inner surface, determined by shear stresses [2]. At some point, these fractures, starting from the inside and outside, reach the other surface or meet and lead to the formation of fragments, also known in the field of ballistics as shrapnel or splinters. At the same time, there is a process of fusion of the fractures in the longitudinal direction, which is why the resulting splinters usually have an elongated and irregular shape but also variable sizes and mass.

Although the process of natural fragmentation is apparently random, the experimental data on the distribution of fragment mass fall into patterns; this is why, over time, various researchers have adopted and justified different statistical laws capable of describing the mass distribution of fragments produced when a shell explodes, with greater or lesser accuracy. Among the first researchers involved in the study of fragmentation is Justrow, who in the 1920’s have adopted an empirical model for military applications [3]. An approach with more mathematical consistency is used by Rosin and Rammler [4], who in 1933, dealing with the subject of coal fragmentation, introduced concepts such as the distribution of fragment sizes in natural fragmentation.

Remarkable research in this field was carried out by Mott during the Second World War [5], who addressed the process of natural fragmentation both for simple, ideal configurations, like rings, but also for projectile bodies or bombs that represent continuous bodies of a form that often cannot be so easily approximated by a single cylindrical body. Developing the research and theoretical observations made by Lienau regarding the fragmentation of a bar [6], Mott initially proposed a cumulative distribution of the fragments number as a function of mass starting from the linear dependence between the logarithm of the cumulative fragments number and the third-order radical of the fragment mass. He later concluded that it is more correct to relate the cumulative number of fragments to the second-order radical of the fragment mass, since most fragments analyzed contained both the internal and external surface of the projectile body [7]. Moreover, Mott deduced a formula that allows the determination of the characteristic mass *µ*, which controls the distribution of the fragments and represents half of the fragments’ average mass. The formula uses as inputs the dimensions of the projectile body and the type of explosive by introducing a material constant known as Mott’s constant.

Subsequent studies and research led to a generalization of the relations proposed by Mott, which is known in the specialized literature as the generalized Mott distribution [7]; it uses a parameter *β* as a power instead of 0.5. The shape parameter *β* is calculated based on the available experimental data. For *β* = 1, the distribution is linearly exponential and is known as the Grady distribution.

To take into account the fragmentation inhomogeneity of real bodies with complex geometry, Grady and Kipp [8] proposed an approach in which the projectile body is divided into two distinct regions. For each of these two regions, the fracturing is considered to be homogeneous. In this way, the projectile fragmentation process is characterized by a bilinear exponential distribution, which is based on three parameters: two average values of the fragment mass corresponding to these two regions, which are independent, and the fractions of the projectile mass corresponding to each region, values that are related to each other, as long as their sum is always 1. Subsequent research has demonstrated that the bilinear exponential distribution is able to reproduce the natural distribution of fragments if they are divided into two categories, large fragments and small fragments [9]. This approach represents a particular case of a statistically inhomogeneous distribution that can be approximated by a combination of linear exponential distributions. In the same way, several generalized Mott distributions can be combined, a combination known as the Weibull hyper distribution [10]. Other researchers suggested a modification of Mott’s formula to take into account the maximum fragment mass [11].

In the last decades, with the increase in computing power, numerical calculation methods have been imposed as an alternative to the traditional means of study. These methods can produce results that are faithful to reality if the material models are correctly chosen and the discretization of the bodies is appropriate [12,13,14]. However, the predetermined expeditious formulas like Mott’s formula do not require a consistent calculation effort nor do they require special computer programming skills by the user, which is why they remain a tool that is still used as a benchmark in design calculations.

In a more recent theoretical and experimental research [15], the fragmentation of a metal cylinder was studied for a setup configuration in which the metal is not in direct contact with the explosive charge, with an inert layer being used to separate them. Starting from the approach proposed by Mott, a modified formula was deduced for the characteristic mass µ. The presence of the inert layer was taken into account by its mass expressed as a percentage of the metal case mass.

Even if the subject of natural fragmentation has been extensively analyzed by numerous researchers, there are no studies dedicated to the derivation of specific formulas for multilayer structures like two or more coaxial metallic cylinders. The present research represents a theoretical study dedicated to the calculation of the natural fragmentation for such configurations, based on the approach proposed by Rotariu [15]. The resulting distribution represents a combination of classical Mott distributions in which the cumulative number of fragments is related to the second-order radical of the fragment mass. In addition, it was also investigated how, for the same total metal mass, the variation in the number of tubes and of their thickness influences the fragments distribution.

## 2. Mathematical Deduction of Fragment Distribution Law

Mott concluded in his reference study [5], based on a careful analysis of the fragments morphology, that the cumulative fragments number distribution resulting from munitions explosions would be described with the following:(1)$$N\left(m\right)=N_{0}e^{-{\left(\frac{m}{\mu }\right)}^{\frac{1}{2}}},$$
where $N_{0}$ is the total number of fragments and *µ* the characteristic mass that represents half of the fragments’ average mass. Associated with this law is the following fragment mass probability density distribution:(2)$$f\left(m\right)=\frac{1}{2\mu }{\left(\frac{m}{\mu }\right)}^{-\frac{1}{2}}e^{-{\left(\frac{m}{\mu }\right)}^{\frac{1}{2}}}.$$

When a cumulative distribution of the fragments number is admitted according to Formula (1), the average mass of fragments $\bar{m}$ and the cumulative fragments mass distribution $M\left(m\right)$ can also be calculated with the following:(3)$$\bar{m}=2\mu =\frac{M_{0}}{N_{0}},$$
(4)$$M\left(m\right)=M_{0}\frac{1}{2}\Gamma \left(3,{\left(\frac{m}{\mu }\right)}^{\frac{1}{2}}\right).$$

It should be noted that in his approach, Mott approximates the body of the shell with a metal cylinder loaded with explosive, Figure 1a, and obtains an intermediate formula in which the characteristic mass μ is expressed as a function of the geometric dimensions of the cylinder at the moment of fragmentation, the average radius r and the thickness t of the wall, the radial velocity at the moment of fragmentation v and the parameters related to the material, the density $\rho_{m}$ and the specific fracture energy W, the latter two being included in a proportionality constant K [5].
(5)$$\sqrt{\mu }=Kr^{\frac{2}{3}}t^{\frac{1}{2}}v^{-\frac{2}{3}}.$$

When an inert intermediate layer lays between the explosive charge and the metal wall, Figure 1b, the structure presents a clear difference from the standard configuration for which Mott proposed his famous equation.

However, it is noted that Formula (5) can also be applied when an inert intermediate layer is present, provided that it is necessary to find how the intermediate layer presence influences the value of the radial velocity of the shell and the geometric dimensions r and t from the moment of fragmentation.

As long as the analyzed configuration preserves the axially symmetrical character, shown in Figure 1b, the same reasoning presented by Grady [7] can be used to express the shell velocity as a function of the specific geometric dimensions, the average radius r and the wall thickness t. The respective reasoning starts from the principle of energy conservation, given by the following relation:(6)$$\frac{1}{2}M_{m}v^{2}\left(r\right)+\frac{1}{2}M_{l}v^{2}\left(r\right)+\omega U\left(r\right)=\omega E.$$

The first term on the left-hand side represents the kinetic energy of the metal body of mass $M_{m}$, the second term represents the kinetic energy of the inert layer of mass $M_{l}$, while the third term on the left-hand side represents the compression energy $\omega U\left(r\right)$ stored in the gases produced upon detonating the charge explosives ω. $U\left(r\right)$ is defined as a specific compression energy $U\left(r\right)$, which, like $v\left(r\right)$, depends on the argument r, the mean radius of the metallic body. The use of an argument indicates that the values of these three energies change as the gases expand and the metallic body is accelerated. The right-hand side represents a fixed value, where E is a material constant equal to the specific compression energy before the gases start to expand. It should be noted that the specific energy E from relation (6) has a similar role to the specific energy defined by Gurney in his calculation of the metallic body velocity [16]. As Grady also observes [7], in relation (6) is neglected the energy consumed with the deformation of solid materials, but above all it, does not discriminate between the kinetic energy of the gases and the internal energy of the gases. Actually, $\omega U\left(r\right)$ represents the sum of these two types of energy. In addition, in relation (6), the simplifying assumption that the inert layer and the case have the same radial velocity $v\left(r\right)$ is admitted.

The mass of the metal body represents a percentage of the total mass of the metal body and the inert layer. This situation is mathematically described by the relation $M_{m}=\varepsilon (M_{m}+M_{l})$. Appling this relation in Formula (6) leads to the following:(7)$$v^{2}\left(r\right)\frac{M_{m}}{\varepsilon \omega }=2E\left(1-\frac{U(r)}{E}\right)$$
or to
(8)$$v^{2}\left(r\right)\frac{d_{m}^{2}-d_{l}^{2}}{d_{e}^{2}}\frac{1}{\varepsilon }=2E\left(1-\frac{U(r)}{E}\right)\frac{\rho_{e}}{\rho_{m}},$$
when the mass ratio is written as a function of the geometric dimensions and density of the metal case $\rho_{m}$ and of the explosive charges $\rho_{e}$, as follows:(9)$$\frac{M_{m}}{\omega }=\frac{d_{m}^{2}-d_{l}^{2}}{d_{e}^{2}}\frac{\rho_{m}}{\rho_{e}}.$$

Using the same assumption proposed by Mott [5] and recognizing the consequence of energy conservation by Grady [7], namely the left-hand side of relation (8) has a constant value for the moment when the metallic body is fragmented, regardless the value of the mass ratio $\frac{M_{m}}{\omega }$, the second-order radical of the right-hand side of the relation becomes a characteristic of the metal/explosive combination, denoted by $u_{0}^{}$ [7].

Under these conditions, we can calculate the metallic body radial velocity at the time of fragmentation with the following relation:(10)$$\frac{1}{v^{2}}=\frac{8}{u_{0}^{2}}\frac{rt}{d_{e}^{2}}\frac{1}{\varepsilon },$$
where $r=\frac{d_{m}+d_{l}}{4}$ is the metallic body mean radius and $t=\frac{d_{m}-d_{l}}{2}$ is the metallic body thickness.

After replacing the term $\frac{1}{v^{2}}$ in relation (5), this results in the following:(11)$$\sqrt{\mu }=K{\left(\frac{1}{u_{0}^{2}}\right)}^{\frac{1}{3}}{\left(\frac{1}{\varepsilon }\right)}^{\frac{1}{3}}\frac{2rt^{\frac{5}{6}}}{d_{e}^{\frac{2}{3}}}.$$

Knowing that the term $K{\left(\frac{1}{u_{0}^{2}}\right)}^{\frac{1}{3}}$ represents Mott’s constant, denoted by C, the relation for the calculation of the characteristic mass in the presence of an inert intermediate layer of mass ${\;M}_{l}={\;M}_{m}\frac{1-\varepsilon }{\varepsilon }$ is given by the following:(12)$$\sqrt{\mu }=C{\left(\frac{1}{\varepsilon }\right)}^{\frac{1}{3}}\frac{2rt^{\frac{5}{6}}}{d_{e}^{\frac{2}{3}}}.$$

Since the density and the length of the metallic body are considered to be maintained constant during the radial expansion of the body, we obtain rt=const. According to Mott in this situation, Formula (12) can be used for the initial geometric data for the radius and wall thickness of the metal case [5].

It should be noted that when the intermediate layer is made of the same material as the outer cylinder, basically two steel cylinders arranged coaxially, and the accepted hypothesis is that the fragmentation occurs simultaneously in both bodies, and the fracturing in one body does not influence the fracturing in the second body; two characteristic masses can be calculated, one for each cylinder: (13)$$\sqrt{\mu_{1}}=C{\left(\frac{1}{\varepsilon_{1}}\right)}^{\frac{1}{3}}\frac{2r_{1}t_{1}^{\frac{5}{6}}}{d_{e}^{\frac{2}{3}}}$$
and
(14)$$\sqrt{\mu_{2}}=C{\left(\frac{1}{\varepsilon_{2}}\right)}^{\frac{1}{3}}\frac{2r_{2}t_{2}^{\frac{5}{6}}}{d_{e}^{\frac{2}{3}}},$$
and the cumulative fragments number distribution would be described by the following:(15)$$N\left(m\right)=\frac{M_{1}}{2\mu_{1}}e^{-{\left(\frac{m}{\mu_{1}}\right)}^{\frac{1}{2}}}+\frac{M_{2}}{2\mu_{2}}e^{-{\left(\frac{m}{\mu_{2}}\right)}^{\frac{1}{2}}},$$
which, in a generalized form that is valid for *n* coaxial cylinders, becomes the following:(16)$$N\left(m\right)=\sum_{i=1}^{n}\frac{M_{i}}{2\mu_{i}}e^{-{\left(\frac{m}{\mu_{i}}\right)}^{\frac{1}{2}}},$$
while the cumulative fragments mass distribution $M\left(m\right)$ becomes the following:(17)$$M\left(m\right)=\sum_{i=1}^{n}M_{i}\frac{1}{2}\Gamma \left(3,{\left(\frac{m}{\mu_{i}}\right)}^{\frac{1}{2}}\right).$$

## 3. Critical Analysis of the Simplifying Assumptions

### 3.1. Equal Radial Velocity for All Cylinders

In obtaining Formulas (16) and (17), an additional hypothesis was used, compared to those used by Mott in his research, by assuming that the radial velocity is the same at every point through the thickness of the cylinders’ walls. The increase in radius under the action of the gases produced by the detonation is accompanied by a reduction in the wall thickness, a fact that can only occur if there is a variation in the radial velocity through the wall thickness, even when a single cylinder is used. To evaluate the differences between the velocity of the inner surface $v_{in}$ and the velocity of the outer surface $v_{ex}$ of such a cylindrical body or a set of cylindrical bodies, the relationship $t=\frac{d_{m}-d_{e}}{2}$ is derived with time, and accepting that rt=const, the following relation is obtained:(18)$$v_{ex}-v_{in}=-\frac{t}{r}v_{med},$$
where $v_{med}$ is the velocity corresponding to the points on the mean radius. A negative value of the $v_{ex}-v_{in}$ difference is the mathematical expression of the wall thinning process, which occurs simultaneously with the radial expansion of the metal body when the density and the length of the metallic body are considered to be maintained constant, and the effect of initial strong shock and rarefaction waves that act on the cylinder is neglected.

Using relation (9) particularized for $d_{e}=d_{l}$ and the expressions of the initial values for the mean radius r and thickness t, it is shown that the initial difference between the velocity of the inner surface $v_{in}$ and the velocity of the outer surface $v_{ex}$ can be expressed as a percentage of the velocity of the points on the mean radius $v_{med}$ with the following:(19)$$\frac{v_{in}-v_{ex}}{v_{med}}=2\frac{\sqrt{\frac{M}{\omega }\frac{\rho_{e}}{\rho_{m}}+1}-1}{\sqrt{\frac{M}{\omega }\frac{\rho_{e}}{\rho_{m}}+1}+1}.$$

It can be seen from relation (19) that this difference depends only on the ratio $\frac{M}{\omega }$ when the same explosive is used. Furthermore, relation (18) shows that as the cylinder grows in diameter due to the action of the gases, this difference is reduced to a value that is proportional to factor $k^{2}$. k represents the ratio between the initial mean radius and the current one.

In Figure 2 are reproduced both the initial ratio $\frac{v_{in}-v_{ex}}{v_{med}}$ for k=1, and the ratio $\frac{v_{in}-v_{ex}}{v_{med}}$ for k=0.5. k=0.5 was chosen because it was considered that fragmentation of the projectiles occurs when the diameter reaches a double value compared to the initial one [17]. The differences tend to increase with the value of $\frac{M}{\omega }$, but it was noted that they have a reasonable value for k=0.5. For example, for $\frac{M}{\omega }=5$, this difference is 8.7% of the $v_{med}$ velocity. Moreover, in relation (6), we work with the average velocities of the bodies, so the difference $v_{in}-v_{ex}$ is clearly greater than the difference between the average velocities of the bodies, which is why the hypothesis used is viable. An additional analysis of the validity of the relationships between the inner and outer surface velocities is given in Section 3.4, based on numerical simulations.

### 3.2. The Existence of the $u_{0}$ Characteristic

This hypothesis is basically based on the assumption of a constant value of the specific compression energy $U\left(r\right)$ at the moment of fragmentation. In support of this hypothesis there are other indirect observations, such as those related to the blast wave effect of ammunitions. Research has shown that the explosive charges placed inside of an ammunition have a weaker blast wave effect than when they are detonated without a metal casing, an effect comparable to that of a smaller charge called the equivalent bare charge, $\omega_{eq}$. Many formulas proposed over time for determining the $\omega_{eq}$ are in fact functions of a single argument, namely the $\frac{M}{\omega }$ ratio, but the following incorporates two other constants, h and α [17,18,19,20,21,22]:(20)$$\frac{\omega_{eq}}{\omega }=(1-\alpha )+\frac{\alpha }{1+\frac{hM}{\omega }}.$$

The constant h defines the distribution of the detonation gases inside the projectile body during the expansion process. The value 1 corresponds to a uniform distribution, and the value 2 corresponds to a non-uniform distribution. The constant α varies between 0 and 1, and represents the percentage of the gases’ total energy that has been transformed into the kinetic energy of the gases and of the projectile body at the time of fragmentation. As a rule, for projectiles with thick walls, the value of α is taken as 0.8 [20]; for pre-fragmented ammunition, in which a weaker confinement is achieved, α is equal to 0.4 [20]. Figure 3 shows the evolution of the ratio $\frac{\omega_{eq}}{\omega }$ for the case where α=0.6 and h=2, a combination that corresponds to the formula proposed by Warren [22].

As can be seen in Figure 3, for variations in the $\frac{M}{\omega }$ ratio between 3 and 10, limits between which most explosive munitions fall, the $\frac{\omega_{eq}}{\omega }$ ratio varies very little, between 0.48 and 0.43. Consequently, there is also a variation within tight limits of the energy that remained stored in the gases able to generate blast waves in the air. It is obvious that this energy quantity is in a direct proportional relationship to the specific compression energy $U\left(r\right)$ used in relation (8). Similar small variations in the ratio $\frac{\omega_{eq}}{\omega }$ are given by Formula (20) when other combinations of the constants h and α are used.

### 3.3. Using the Initial Values of r and t in Relation (12)

According to Mott, since rt=const and the product $rt^{\frac{5}{6}}$ varies slowly as the average radius r increases, Formula (12) uses the initial geometric data for the average radius r and the wall thickness t of the metal case. To better understand this approximation, we can calculate the product $rt^{\frac{5}{6}}$ when the average radius of the body is doubled, a value at which projectile bodies fragment [17]. As can be seen in Figure 4, the product $rt^{\frac{5}{6}}$ increases by only 12.2% compared to the initial value. In Figure 4 is plotted the evolution of the $rt^{\frac{5}{6}}$ product normalized to its initial value (k=1) as a function of the ratio between the current mean radius and the initial one, written as $\frac{1}{k}$.

### 3.4. No Interference between the Cylinders’ Fragmentations Processes

Related to the simultaneous occurrence of the cylinders’ fragmentation, certainly each individual cylinder registers for the same moment of time different average strain values. The highest strain value is always obtained in the inner cylinder. However, it is considered that the cracks start from both the inner and the outer surfaces, which is why it is difficult to specify if there are large time differences between the moments of the cylinders’ fragmentations, and especially if these differences modify in any way the fragmentations of the cylinders.

A more in-depth analysis of these aspects can be carried out based on numerical simulation, a modern tool used successfully in numerous studies dedicated to the fragmentation of bodies subjected to high loads produced by detonation or hypervelocity impacts [23,24,25,26,27]. In this study, the advantage of numerical simulation lies in the possibility to create virtual models that incorporate conditions impossible to achieve from a practical point of view. In this sense, a planar 2D model was built in AUTODYN, in three different versions. In the first version, in an Euler part of a 140 × 140 mm element filled with AIR, an explosive charge of C4 with a diameter of 37 mm was defined. The explosive was surrounded by two cylindrical and coaxial Lagrange parts, each of them with a wall thickness of 3.5 mm and loaded with STEEL 1006; see Figure 5a. These dimensions correspond to a ratio $\frac{M}{\omega }=4.46$.

The materials were chosen from the material database of AUTODYN [28]. For the Lagrange parts, the effective plastic strain equal to 1 was established as the failure criterion, with the option of a stochastic variance of 16, a minimum fail fraction of 0.9 and a random seed, values used in a previous study [29]. The Euler part was meshed by a network of nodes defining 0.5 × 0.5 mm square finite elements. For the circular Lagrange parts, the number of nodes on the radius and circumferential directions were defined in order to obtain a fine mesh and reasonable initial ratio of the quadrilateral elements’ sides. The erosion option for the Lagrange elements was linked to the failure criterion. Both the stochastic variance option and the fine mesh of cylindrical parts contribute to a small percentage of eroded mass during the fragmentation process. To avoid repeated collisions between the cylinders, a situation that occurs when a single central detonation point is defined, other eight concentric detonation circles were defined at distances of 2 mm between them, as seen in the Figure 5. This setting option approximates the specific mode of interaction between the detonation wave and the wall for long cylindrical structures, when detonation initiation occurs at one end.

The combination of the chosen value for the failure threshold and the random distribution of the properties led to a fragmentation process that lasted until the outer cylinder exceeded twice its initial diameter. The remaining steel mass when erosion stops was 81.5% of the initial value, and the AUTODYN automatic option of fragments counting counted 151 fragments. In Figure 6, a sequence in time with a quarter of the model is shown.

In the simulation, the cylinders did not fragment simultaneously; the first cracks appeared in the inner cylinder, and were produced by the shear stresses, contrary to the classical theoretical work findings [30,31]. It can be tempting to try a justification based on the fact that a consistently higher failure threshold was used than the real ones, but in the simulations performed with such values, the first cracks appeared also on the inner surface of the projectile [32]. It should be noted that in the classic works mentioned above was explained that the occurrence of fractures on the inner surface is inhibited in the initial phases, because there are compressive hoop stresses over an inner region of the metallic casing when high pressure is applied at the interior surface. More recent research [33] has shown that the equivalent strain to fracture is in fact a non-monotonic function of the stress triaxiality, which tends to infinity when the triaxiality goes down to −0.333. As long as the failure criterion defined in the simulation has a fixed value, which does not take into account the stresses’ triaxiality, and the inner zone always suffer much higher strains than the outer one, it was expected that in the simulation the cracks should start from the inside.

Moreover, some cracks appearing in the inner cylinder continued in the outer cylinder as well, as can be seen in the areas framed with purple ellipses in Figure 6.

In the second studied variant, see Figure 5b, the failure and erosion options were cancelled for the inner cylinder material, and in the third, the same modification was applied only to the outer cylinder; see Figure 5c. Through these changes, the occurrence of fractures in one of the cylinders was alternatively blocked. In addition, another version of the numerical model was built; see Figure 5d. This time, the cylinders formed a single body. All three modified variants were performed, benefiting from the AUTODYN automatic option of fragments counting a list of the fragments, and their mass was obtained for each simulation after the fragmentation stopped. The total number of fragments for each simulation and a comparative view of the cylinders’ statuses at 32 μs are shown in Figure 7.

The lists of fragments from the modified simulations with the fracturing option canceled for one of the cylinders were cumulated, and the results were compared with the cumulative fragment distribution obtained for the first variant, see Figure 8. The differences between these two distributions show that mutual influences that occur during the fragmentation of the cylinders, such as the cracks that cross both cylinders, do not produce significantly different results compared to the individual fragmentation of the cylinders. This result is in agreement with the assumed hypothesis of independent fragmentation of the cylinders. In the same figure, the results of the simulation with a single body are reproduced.

In Figure 8 are also represented two generalized Mott distributions. The first one, which approximates the result obtained for the single-body variant, was obtained through mathematical regression; the value of the parameter *β* was 2, and the value of the scale parameter was 96 mg. A total of 74 fragments corresponded to this distribution. The second one, which reproduces the results of the two-body variants, was obtained by halving the value of the scale parameter to 48 mg while the parameter *β* was kept to 2. A total of 148 fragments corresponded to this distribution. It is observed that the simple halving of the scale parameter from the single body case allows a good approximation of the distribution of the fragments in the two-body case when the distribution obtained for the single-body case is used.

To complete the analysis provided in Section 3.1, in the simulation with two separate cylinders, two gauges were defined; one was on the inner surface of the inner cylinder, and the other one on the outer surface of the outer cylinder. The evolutions of the density and of the radial velocity on these two gauges were calculated and plotted in Figure 9. As can be seen, the first period, marked by the velocities’ strong oscillations and an important increase in density on the inner surface, corresponds to the occurrence and the propagation of shock and rarefaction waves back and forth through the cylinders. In the second period, the density returns to values close to the initial value and the velocities maintain an increasing trend, with the speed of the inner surface being higher than the speed of the outer surface; this situation corresponds to the relationships established in Section 3.1.

In the same figure, the values of the average radial speeds of the cylinders are shown. Once the oscillations fade, these two curves fall within the range determined by the speed of the inner surface and that of the outer surface. At the end of the plotted period, the difference $v_{in}-v_{ex}$ is 8.7% of the $v_{med}$ velocity, and the difference between the radial velocities of the cylinders is 3.3%. These results show that when the cylinders double their size, the value at which the cylinders are expected to fragment [17], the conditions used in the derivation of relations (18) and (19) are fulfilled.

## 4. Case Study on Structures with Two and Three Coaxial Cylinders

To evaluate how the existence of two or more coaxial cylinders influences the fragmentation when a centrally placed explosive charge detonates, the following configuration inspired from a real configuration used in experiments [15] was imagined: a central charge made of RDX with a diameter of 37 mm and a length of 65 mm, which is encapsulated in a steel case of the same length, and with a total wall thickness of 14 mm. In addition to the variant in which the metal case is made from a single body, five other variants were taken into account, three with two cylinders and two with three coaxial cylinders; see Figure 10.

Two variants were imagined, starting from the idea of equal thicknesses of the cylinders’ walls, and another two from the idea of equal masses of the cylinders. The last version takes into account the existence of an thin inner cylinder and an thick outer one.

Table 1 shows the relevant data of all six configurations. For all configurations, the scale parameters of the fragments mass were calculated. In the calculations, we used the Mott’s constant that corresponds to RDX, namely 2.597 kg^1/2^ m^−7/6^. The results of the calculations are presented in Table 2.

Figure 11 shows the cumulative fragment distributions for all six configurations. As expected, the number of fragments increases with the number of cylinders, and it is noted that there are no noticeable differences between the variants with equal thicknesses and those with equal mass. Moreover, the analysis of the total number of fragments shows that practically there is a linear relationship between the total number of fragments and the number of cylinders. For this reason, on the same graph are also plotted the cumulative fragment number distributions defined by the following:(21)$$N_{n}\left(m\right)=nN_{0}e^{-{\left(\frac{nm}{\mu }\right)}^{\frac{1}{2}}},$$
where the data from the first line of Table 2 are used for the total number of fragments $N_{0}$ and the scale parameter μ, and for *n*, the values 2 and 3.

The curves drawn for relation (21) overlap with those drawn for configurations 2 and 3 when n=2, and with those drawn for configurations 4 and 5 when n=3. For this reason, it can be concluded that when the coaxial cylinders have comparable dimensions, the distribution of fragments can be estimated simply starting from the calculations performed for a single whole cylinder, and then Formula (21) is applied for the specific number of cylinders. This conclusion is also supported by the results of the simulations presented in Section 3.4, where the two generalized Mott distributions used for the approximation of simulations results use as the total number of fragments the value 74 for the single-body case, and double, 148, for the two-body case.

At the same time, it is observed that when there are major differences between the cylinders, as in configuration no. 6, expeditious Formula (21) cannot be used, and it is necessary to go through the entire calculation algorithm that leads to Formula (16).

The use of Formula (21) allows for a quick calculation of the ratio between the number of fragments of mass higher than *m* generated by *n* tubes $N_{n}\left(m\right)$, and the number of fragments of mass higher than *m* generated from a single tube $N\left(m\right)$ with the following formula:(22)$$\frac{N_{n}\left(m\right)}{N\left(m\right)}=ne^{{\left(\frac{m}{\mu }\right)}^{\frac{1}{2}}-{\left(\frac{nm}{\mu }\right)}^{\frac{1}{2}}}=ne^{{\left(\frac{m}{\mu }\right)}^{\frac{1}{2}}(1-\sqrt{n})}.$$

Accepting Formula (21), one can quickly identify the value of the mass *m_e_* for which the number of fragments with a mass higher than *m_e_* generated by *n* cylinders becomes greater than the number of fragments generated from a single cylinder by defining the following equality:(23)$$ne^{{\left(\frac{m_{e}}{\mu }\right)}^{\frac{1}{2}}(1-\sqrt{n})}=1,$$
that leads to
(24)$$m_{e}=\mu {\left(\frac{\ln\left(n\right)}{\sqrt{n}-1}\right)}^{2}.$$

As a miscellaneous fact, analyzing the above formula, it can be seen that if four or more cylinders are used, the $m_{e}$ value drops below 2*μ*, the value that corresponds to the average mass for a single cylinder.

From the ballistic efficiency point of view, a solution involving several cylindrical bodies can be judged starting with the number of fragments formed that have a mass greater than the mass $m_{min}$ considered to be the minimum mass for which the fragment has relevance in terminal ballistics. A higher $\frac{N_{n}\left(m_{min}\right)}{N\left(mm_{min}\right)}$ ratio means a better fragmentation, provided it is above unity. For example, if the studied cases admit a minimum mass $m_{min}$ of 0.5 g, the one-cylinder configuration produces a number of 233 fragments with a mass greater than 0.5 g; the two-cylinder configurations provides an increase of 13.7% because they produce 265 fragments with a mass greater than 0.5 g, and the three-cylinder configurations only provide an increase of 11.1% because they produce 259 fragments with a mass greater than 0.5 g.

## 5. Conclusions

(1)The fragmentation of several coaxial cylindrical bodies under the action of gases produced by detonation can be theoretically approached by adapting the theoretical model proposed by Mott for the fragmentation of a single cylindrical body. For this purpose, two additional hypotheses were used: equal radial velocity for all cylinders, and no interference between the fragmentation processes of the cylinders.(2)Thus, the formulas obtained show that in the calculation of the fragments’ mass scale parameters for each individual body, the influence of the other bodies is made through the dimensionless term ε that represents the ratio between the mass of the body in question and the total mass of all bodies.(3)For all simplifying hypotheses used in the development of the model, a justification analysis was presented.(4)For the “no interference” hypothesis, the results of a series of 2D planar numerical simulations were used. It was highlighted that the fragmentation process of two coaxial cylinders is not simultaneous, and there are some mutual influences; however, it can be approximated by accumulating the results obtained for each cylinder fragmented separately when the influence of the other cylinder fragmentation is cancelled by eliminating the failure and erosion criteria from the assigned material model. Such a result supports the use of the assumed hypothesis of the cylinders’ independent fragmentation.(5)Even if in the simulations the stochastic variance option was used in the plastic strain failure criterion to obtain an uneven distribution of the failure threshold, this approach fails to take in account the effect of the stress triaxiality on the value of the equivalent strain to fracture.(6)Theoretical calculations for configurations with two or three cylindrical bodies that have equal masses or equal wall thicknesses showed that the fragment mass distributions can be approximated by the generalized relation (21) when there are no large discrepancies between the cylindrical bodies in terms of their dimensions.(7)When there are major differences between the cylinders, Formula (16) is used to calculate the fragments’ mass distribution.(8)Although it does not fall into the category of controlled fragmentation solutions, in the absence of elements such as notches on the surfaces, which is why it does not have the same potential, the natural fragmentation of several coaxial cylinders can bring benefits from the point of view of terminal ballistics. Such a solution is justified to be used to the detriment of the single-body solution when it leads to a higher number of fragments that have a mass greater than $m_{min}$, the minimum mass for which the fragment is relevant.

## Figures and Tables

**Figure 1 materials-16-05783-f001:**
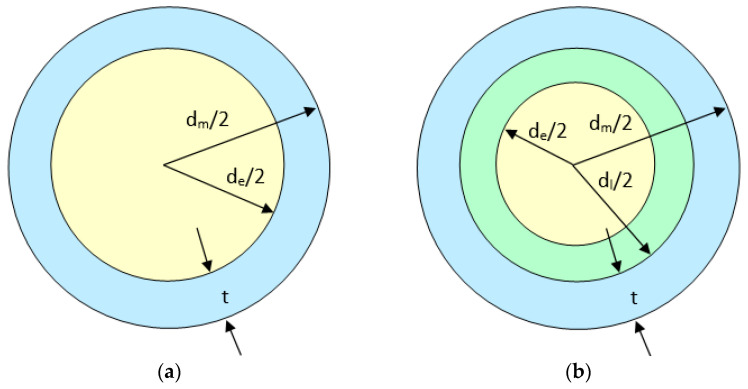
Geometric data required for the calculation of the fragment mean mass based on Mott formula (**a**) vs. geometric data specific to a configuration with an inert interlayer (**b**) [15].

**Figure 2 materials-16-05783-f002:**
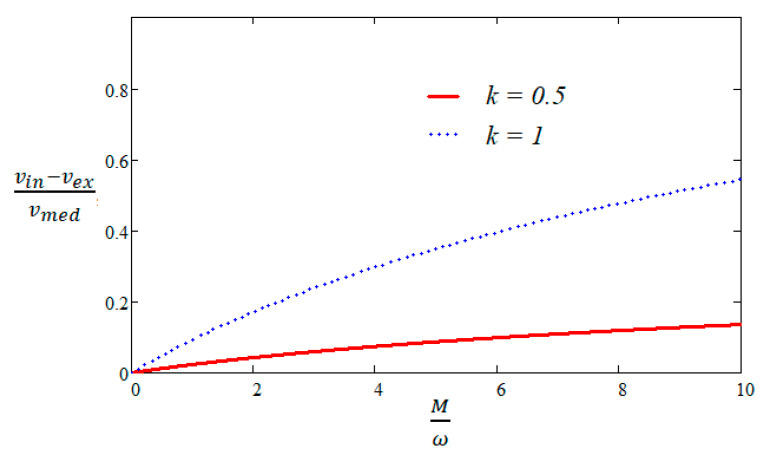
$\frac{v_{in}-v_{ex}}{v_{med}}$ ratio as a function of $\frac{M}{\omega }$.

**Figure 3 materials-16-05783-f003:**
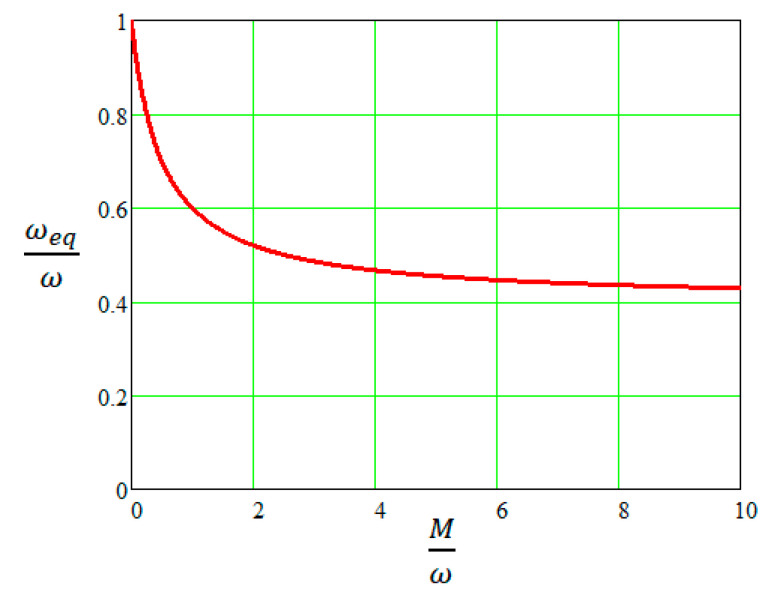
The $\frac{\omega_{eq}}{\omega }$ ratio as a function of $\frac{M}{\omega }$, for α=0.6 and h=2.

**Figure 4 materials-16-05783-f004:**
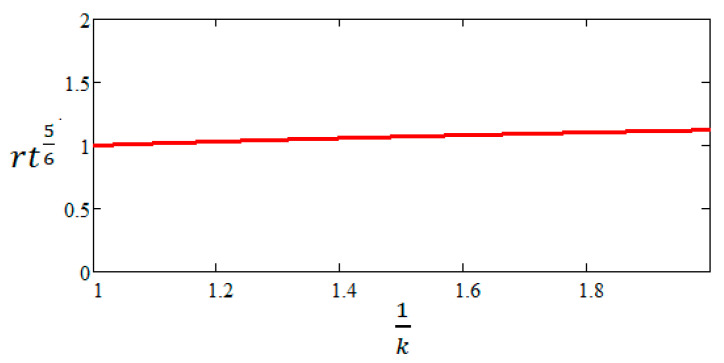
Evolution of the $rt^{\frac{5}{6}}$ product as a function of $\frac{1}{k}$.

**Figure 5 materials-16-05783-f005:**
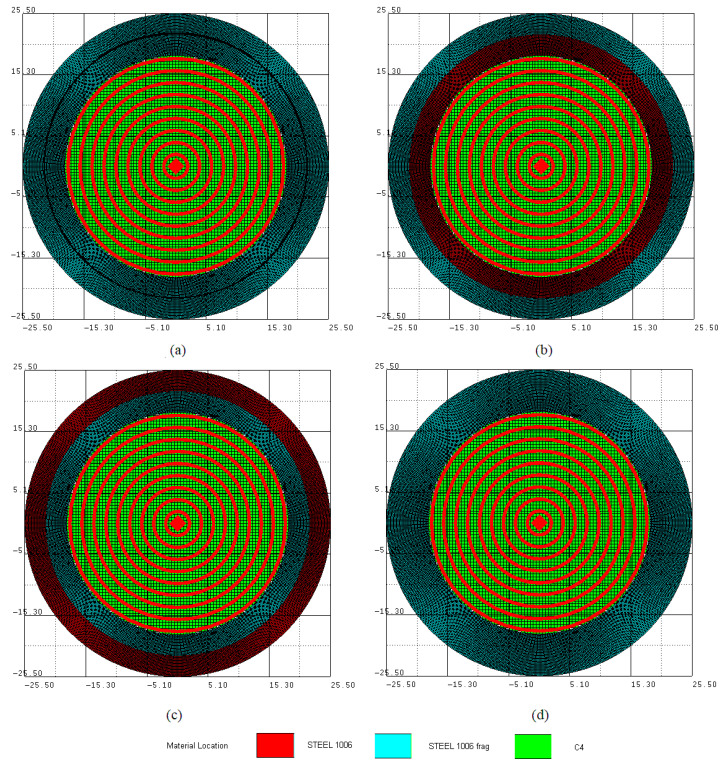
Frontal view of the 2D planar models: (**a**) two coaxial cylinders; (**b**) the erosion option canceled for the inner cylinder; (**c**) the erosion option canceled for the outer cylinder; (**d**) a single cylindrical body.

**Figure 6 materials-16-05783-f006:**
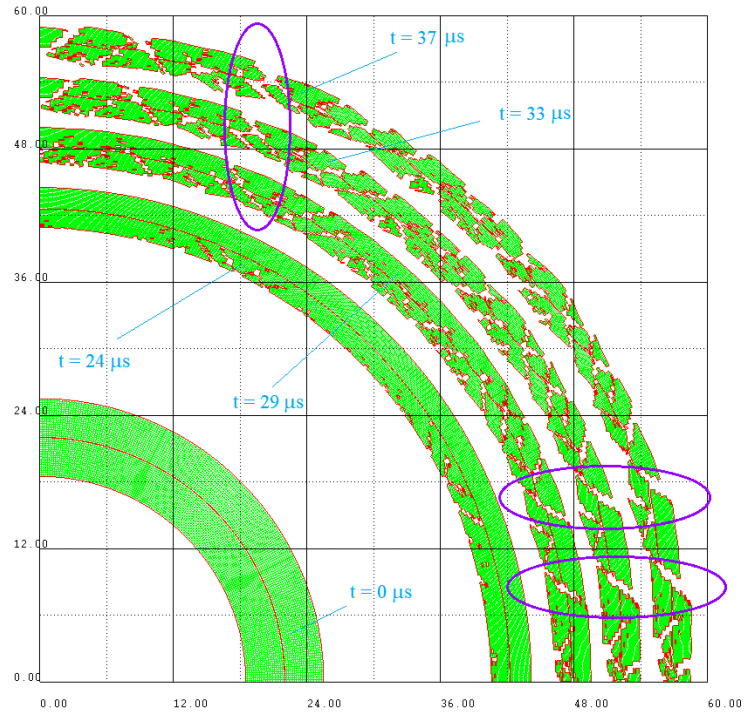
Detail with the successive stages of cylinders fragmentation at 24, 29, 33 and 37 μs from the moment of detonation. The purple ellipses mark examples of cracks that cross both cylinders.

**Figure 7 materials-16-05783-f007:**
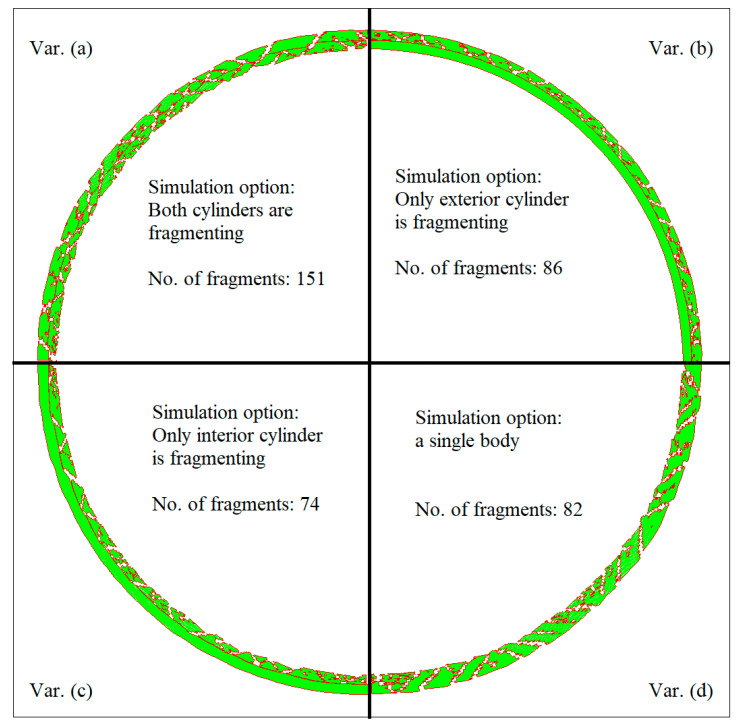
Detail with cylinders’ statuses at 32 μs from the moment of detonation for all variants: (**a**) two coaxial cylinders; (**b**) the erosion option canceled for the inner cylinder; (**c**) the erosion option canceled for the outer cylinder; (**d**) a single cylindrical body.

**Figure 8 materials-16-05783-f008:**
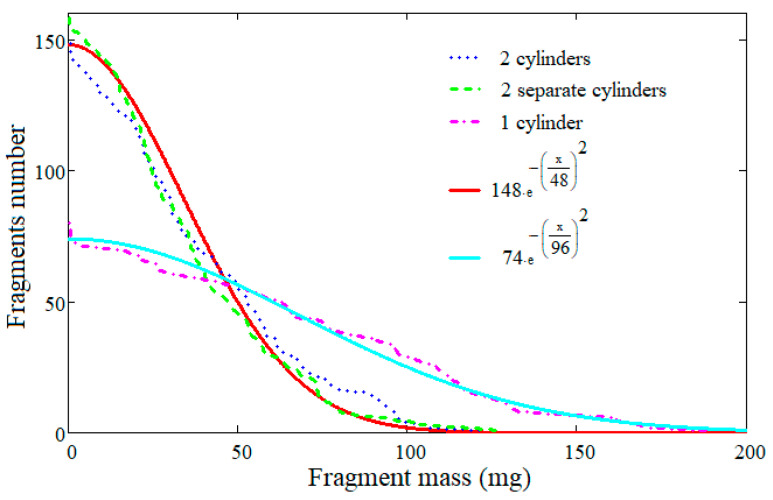
Cumulative fragments number distributions.

**Figure 9 materials-16-05783-f009:**
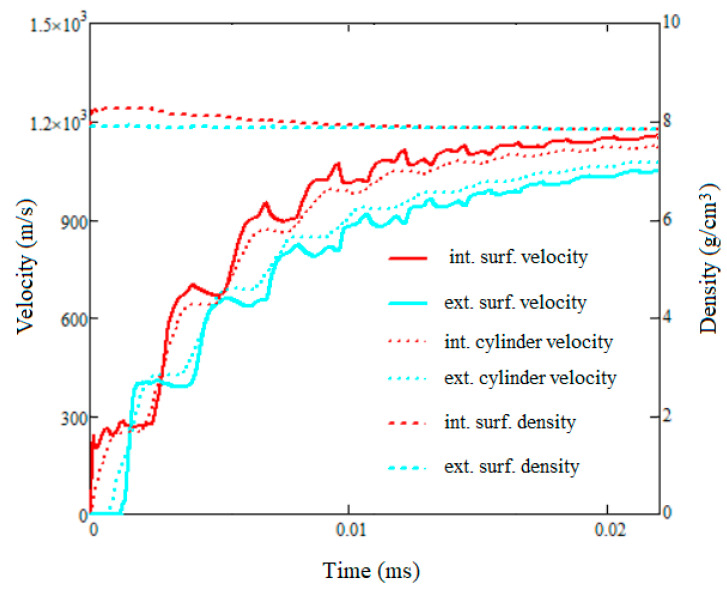
Velocities and densities evolution during cylinder expansions.

**Figure 10 materials-16-05783-f010:**
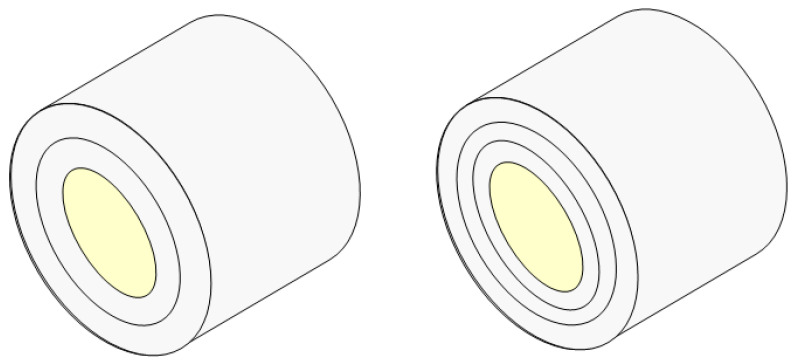
Sketch of the two and three coaxial cylinders configurations.

**Figure 11 materials-16-05783-f011:**
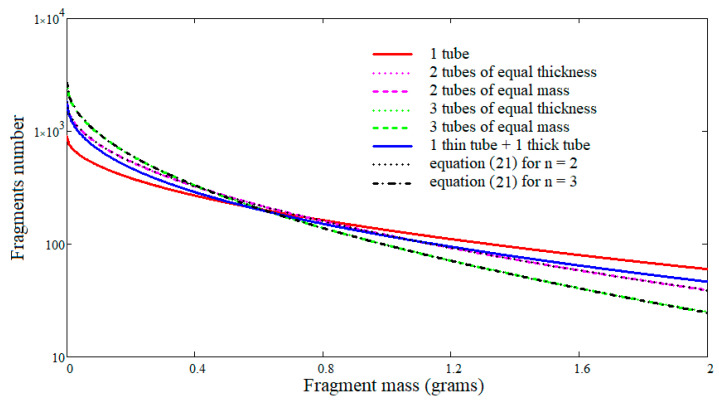
Cumulative fragment number distributions for the defined configurations.

**Table 1 materials-16-05783-t001:** Defined configurations and the main dimensional values.

No.	No. of Cylin.	Tip Conf.	Mean Diam. Cyl. No. 1 [mm]	Wall Thickness Cyl. No. 1 [mm]	MASS Ratio Cyl. No. 1	Mean Diam. Cyl. No. 2 [mm]	Wall Thickness Cyl. No. 2 [mm]	Mass Ratio Cyl. No. 2	Mean Diam. Cyl. No. 3 [mm]	Wall Thickness Cyl. No. 3 [mm]	Mass Ratio Cyl. No. 3
1	1 cyl.	Classic	44	7	-	-	-	-	-	-	-
2	2 cyl.	Eq. thick.	40.5	3.5	0.46	47.5	3.5	0.54	-	-	-
3	2 cyl.	Eq. mass	40.78	3.78	0.5	47.78	3.22	0.5	-	-	-
4	3 cyl.	Eq. thick.	39.33	2.33	0.298	44	2.33	0.333	48.67	2.33	0.369
5	3 cyl.	Eq. mass	39.59	2.59	0.333	44.49	2.315	0.333	48.90	2.095	0.334
6	2 cyl.	Thin + thick	38.5	1.5	0.188	45.5	5.5	0.812	-	-	-

**Table 2 materials-16-05783-t002:** The scale parameters μ and the total numbers of fragments.

No.	No. of Cylinders	Configuration Type	$\mu_{1}$	$\mu_{2}$	$\mu_{3}$	The Total Number of Fragments
1	1 cylinder	Classic	2.713 × 10^−4^ kg	-	-	907
2	2 cylinders	Equal thickness	1.215 × 10^−4^ kg	1.502 × 10^−4^ kg	-	1815
3	2 cylinders	Equal mass	1.325 × 10^−4^ kg	1.392 × 10^−4^ kg	-	1812
4	3 cylinders	Equal thickness	7.787 × 10^−5^ kg	9.046 × 10^−5^ kg	1.034 × 10^−4^ kg	2724
5	3 cylinders	Equal mass	8.712 × 10^−5^ kg	9.126 × 10^−5^ kg	9.335 × 10^−5^ kg	2717
6	2 cylinders	Thin + thick	4.865 × 10^−5^ kg	2.229 × 10^−4^ kg	-	1844

## Data Availability

No new data were created or analyzed in this study. Data sharing is not applicable to this article.

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
