# Peer review of "The Application of Mott’s Distribution in the Fragmentation of Steel Coaxial Cylinders"

_materials, 2023, doi:10.3390/ma16175783_

Round 1

Reviewer 1 Report

The fragmentation characteristics of the natural fragmentation of cylindrical structures with metal cylinders are studied in the manuscript. Especially Mott's Distribution in the Fragmentation of Steel Coaxial Cylinders, and the fragmentation characteristics of two and three coaxial cylinders are analyzed. It is meaningful work, but there are some problems to be further improved.

1. The research is focused on the theoretical study. The results mainly come from the empirical relations and the numerical simulations. Thus, it is suggested to further verify the accuracy of the calculation model in this manuscript. Try to find the data by performing relevant tests or the data in literatures.

2. From Formula 18, Formula 19 and Figure 2, it is shown that the velocity of the inner surface vin is greater than the velocity of the outer surface vex. Why is it not vinvex, please provide the essential support and give the explanation in the text.

3. For Section 3.4, please add the material model, especially the value of the failure and erosion model parameters of the casing. It is suggested to show different simulation results of the four typical cases shown in Figure 5.

4. Briefly explain the expression in Figure 7, how to get the value'148,48,74,96', which formulas are associated with the above ones?

5.For section 4, suggested to increase the numerical model for different working conditions, and verify the accuracy of the calculation model by comparing the results of theoretical calculation and numerical simulation.

The accuracy of the paper writing needs to be further checked. For example, lines 420 and 424 are both numbered 5, which is an obvious writing error.

It is suggested to further refine the conclusions.

Author Response

Dear Colleague,

first of all, I want to thank you for agreeing to analyze the paper written by our team. Also, thank you for the constructive comments related to the structure and its content, which we took into account in the revised form that was uploaded again on the platform. Next, I present to you how we treated each of your observations.

  1. The research is focused on the theoretical study. The results mainly come from the empirical relations and the numerical simulations. Thus, it is suggested to further verify the accuracy of the calculation model in this manuscript. Try to find the data by performing relevant tests or the data in literatures.

As we stated and you also well observed, the study is a theoretical one and aims to adapt the model proposed by Mott for several bodies arranged concentrically. For this reason, the calculation model follows Mott's reasoning to which an additional hypothesis is added: the fragmentation of bodies occurs simultaneous and independently. This hypothesis is analyzed with the help of a simulation, the results showing that the mutual influence of the cylinder fragmentation processes is reduced. Regarding the experimental data, unfortunately there are no relevant data regarding such multi-body configurations, except for those presented by our team in another work mentioned in the introduction and bibliography. But there the intermediate layer did not have the consistency of a solid body, being made of aluminum foil. In the future, we will perform small-scale experiments with real explosives as part of a PhD thesis work and we will have the opportunity to discuss the degree of fidelity of this theoretical model in a future paper.

  1. From Formula 18, Formula 19 and Figure 2, it is shown that the velocity of the inner surface vinis greater than the velocity of the outer surface vex.Why is it not vinvex, please provide the essential support and give the explanation in the text.

The wall thinning process can take place only if the the inner surface vinis greater than the velocity of the outer surface vex. The following explanations were added to paper text.

“Since the density and the length of the metallic body are considered to be maintained constant during the radial expansion of the body, we get ..”

“A negative value of the  difference is the mathematical expression of the wall thinning process, which occurs simultaneously with the radial expansion of the metal body.”

  1. For Section 3.4, please add the material model, especially the value of the failure and erosion model parameters of the casing. It is suggested to show different simulation results of the four typical cases shown in Figure 5.

The vital information regarding how the failure and erosion conditions were defined are present in the text of the paper.

“For Lagrange parts, the effective plastic strain equal to 1 was established as the failure criterion, with the option of stochastic variance of 16, a minimum fail fraction of 0.9 and a random seed, values ​​used in some previous work [23].”

“The erosion option for the Lagrange elements was linked to the failure criterion.”

Some new observations related to the simulation results were added to the paper text. Also was added a new figure with cylinders status at 32 μs from the moment of detonation for all variants

  1. Briefly explain the expression in Figure 7, how to get the value'148,48,74,96', which formulas are associated with the above ones?

The following explanation was added to paper text replacing the old short phrase.

“In the same figure, two generalized Mott distributions are represented. The first one, that approximates the result obtained for the single-body variant, was obtained through mathematical regression, the value of the parameter b being 2 and the value of the scale parameter being 96 mg. A total number of 74 fragments correspond to this distribution. The second one, that reproduces the results of the two-body variants, is obtain by halving the value of the scale parameter to 48 mg while the parameter b was kept to 2. A total number of 148 fragments correspond to this distribution. It is observed that the simple halving of the scale parameter allows a good approximation of the distribution of the fragments in the two-body case when the distribution obtained for the single-body case is used.”

5.For section 4, suggested to increase the numerical model for different working conditions, and verify the accuracy of the calculation model by comparing the results of theoretical calculation and numerical simulation.

Unfortunately we have not build yet 3D numerical models for studied configurations and building, running and interpreting the data of 3D simulations with a fine mesh, of the order of millions of nodes, requires a long working time of the order of months, but Materials journal requested us that the changes made to the content of the work to be carried out within 10 days of receiving your review. Also, I intend to add such research in a future work in which we will present the results of some real small-scale tests that we intend to carry out in the future as a PhD research topic.

Reviewer 2 Report

As the authors point out, a theory for the dynamic fragmentation of cylindrical tubes due to an internal explosion was first proposed during World War 2 by Nevill Mott. The novelty of the new study is that the authors considered coaxial tubes. It is an entirely theoretical study which would be improved if the authors had performed some experiments to back up their ideas. Nevertheless, I am happy to recommend it for publication so long as the authors attend to the points raised below.

Points for clarification

Line 178. I appreciate that the paper is a theoretical study, but I think it unlikely that in a physical experiment fragmentation will take place simultaneously in the two cylinders and that there will be no influence of one cylinder upon the other.

Line 193. This statement is unclear: do they mean that the radial velocity is the same at every point along the length of the cylinder or that the radial velocity is the same at every point through the thickness of the wall of the cylinder?

Line 196. Do they mean “…radial velocity through the wall thickness…”

Minor points

Line 26. …fragments are produced…

Line 46. In the text, reference [1] is stated to be by Justrow and Yulovsky. But in the bibliography, reference [1] is the book by Cranz on internal ballistics.

Line 55. The name of the author of reference [4] is given incorrectly: it should be Lienau.

Line 298. ...the shared stresses...

Line 413. …used in the development of…

Line 422. …fails to take into account…

References

It would be easier for the readers to find reports if report numbers are given. I give this extra information in the list below.

N.F. Mott's 1943 report was reissued in the book edited by D. Grady in 2006. Also Mott first published his ideas on fragmentation in a paper published by the Royal Society of London in 1947 (details given below).

Lienau's 1936 paper was published in three parts (details given below).

Fisher, E.M. 1953. The effect of the steel case on the air blast from high explosives. NAVORD Report 2753, US Naval Ordnance Laboratory, White Oak, MD.

Gurney, R. 1943. The initial velocities of fragments from bombs, shells, and grenades. BRL Report number 405, Ballistics Research Laboratory, Aberdeen, MD.

Lienau, C.C. Random fracture of a brittle solid. J. Franklin Inst. 1936, 221, 485-494, 673-686, 769-787.

Mott, N.F. 1943. Fragmentation of HE shells: A theoretical formula for the distribution of weights of fragments Rep. no. AC3642, Ministry of Supply, London.

Mott, N.F. Fragmentation of shell cases. Proc. R. Soc. Lond. A 1947, 189, 300-308.

Mott, N.F. 2006. Fragmentation of H.E. shells: A theoretical formula for the distribution of weights of fragments. In: D.E. Grady (Ed.), Fragmentation of Rings and Shells: The Legacy of N.F. Mott. Springer, Berlin, pp. 227-241.

Taylor, G.I. 1963. The fragmentation of tubular bombs. In: G.K. Batchelor (Ed.), The Scientific Papers of G.I. Taylor. Vol. 3. Cambridge University Press, Cambridge, pp. 387-390.

Quality of the English OK.

Author Response

Dear Colleague,

first of all, I want to thank you for agreeing to analyze the paper written by our team. Also, thank you for the constructive comments related to the structure and its content, which we took into account in the revised form that was uploaded again on the platform. Next, I present to you how we treated each of your observations.

Points for clarification

Line 178. I appreciate that the paper is a theoretical study, but I think it unlikely that in a physical experiment fragmentation will take place simultaneously in the two cylinders and that there will be no influence of one cylinder upon the other.

The hypothesis was analyzed in section 3.4 using numerical simulation as a study tool. A configuration with two cylinders was built in several variants, see Figure 5. The simulation for the normal variant shows that there is no complete lack of influence of the fragmentation processes, see the fractures that cross both cylinders. However, by combining the fragmentation results obtained in the additional studied variants, in which the fragmentation of one cylinder was blocked, and comparing them with the result obtained for the initial variant, it was shown that the mutual influence is not so important. Moreover, it was shown that the results of the simulation with this two cylinders can be easily approximated starting from the results of the simulation with a single cylinder of equivalent mass, by simply halving the scaling parameter from Mott's generalized law. This last result agrees with the observations from the section 4 for the configurations with cylinders of equal thickness.

The text of section 3.4 has been modified compared to the initial version, to nuance the conclusion regarding this hypothesis. Was introduced references to the existence of mutual influence, see the fractures that cross both cylinders and  that the cylinders do not fragment simultaneously. Figure 6 was also modified and a new figure was introduced.

In the conclusion section we wrote the following

“(4) For the “no interference” hypothesis the results of a series of 2D planar numerical simulations were used. It was highlighted that the fragmentation process of two coaxial cylinders is not simultaneous and there are some mutual influences, but it can be approximated by cumulating the results obtained for each cylinder fragmented separately when the influence of the other cylinder fragmentation is cancelled by eliminating the failure and erosion criteria from the assigned material model. Such a result supports the use of the assumed hypothesis of the cylinders independent fragmentation.

Line 193. This statement is unclear: do they mean that the radial velocity is the same at every point along the length of the cylinder or that the radial velocity is the same at every point through the thickness of the wall of the cylinder?

We have clarified the meaning of the text with the following change

“the radial velocity is the same at every point through the thickness of the cylinders walls”

Line 196. Do they mean “…radial velocity through the wall thickness…”

We have made the modification “of the radial velocity through the wall thickness”

Minor points

Line 26. …fragments are produced…

We  made the correction

Line 46. In the text, reference [1] is stated to be by Justrow and Yulovsky. But in the bibliography, reference [1] is the book by Cranz on internal ballistics.

We have change the reference and the text is now “Among the first researchers involved in the study of fragmentation is Justrow, who in the 1920’s have adopted an empirical model for military applications [3]”

Line 55. The name of the author of reference [4] is given incorrectly: it should be Lienau.

We made the correction.

Line 298. ...the shared stresses...

We made the correction.

Line 413. …used in the development of…

We made the correction.

Line 422. …fails to take into account…

We made the correction.

References

It would be easier for the readers to find reports if report numbers are given. I give this extra information in the list below.

We had added in the references the numbers of the reports.

Reviewer 3 Report

The present work is theoretical in nature and studies the application of Mott’s hypotheses to model the fragmentation process of concentric metal cylinders undergoing explosive detonation. The development of the model, equations and various constants is thorough and in detail. The reviewer insists that the author revisit some of the assumptions and spend some more time either justifying these further or providing an explanation as to how even after the incorporation of these assumptions, the modeling results are meaningful.  

1.       Line 18: Sentence does not make sense. Do the authors mean that can be used?

2.       Lines 23-40: This is a lot of useful information. Some citations to academic work or other articles would be useful.

3.       Line 98: Who does his refer to here?

4.       Lines 159-162: This is not clear. Which assumption by Mott are the authors referring to? Please define time of metallic body fragmentation. Why is the characteristic term denoted by u_0^2?

5.       Line 174: This study appears to be relying on a lot of assumptions from the cited work from Mott. Why is rt constant? Please provide an explanation and comment on if it is justified. And if it is indeed constant, why does that mean that the initial geometric data for the radius and wall thickness of the metal case will be used?

6.       Line 176-180: These are a lot of assumptions. Some of which don’t seem like they can be justified. Can the authors comment on these assumptions? It looks highly unlikely that the fracturing of one body does not influence the fracturing of the second body, especially if they are in such proximity?

7.       Lines 192-194: Another assumption, can this be further justified?

8.       Line 424: The numbering is repeated at 5.

Author Response

Dear Colleague,

first of all, I want to thank you for agreeing to analyze the paper written by our team. Also, thank you for the constructive comments related to the structure and its content, which we took into account in the revised form that was uploaded again on the platform. Next, I present to you how we treated each of your observations.

  1. Line 18: Sentence does not make sense. Do the authors mean that can be used?

We have changed the text as follows

“The results obtained for the structures with two and three cylinders, with equal masses or equal wall thicknesses, can be approximated by a modified Mott's distribution formula in which the number of cylinders is used as an additional parameter.”

  1. Lines 23-40: This is a lot of useful information. Some citations to academic work or other articles would be useful.

We had introduced two new citations

  • Kritzinger, H. H. Stuhlmann F. Artillerie und Ballistik in Stichworten, Berlin, Verlag von Julius Springer, 1939

- Mock, W.: Holt, W. H. Fragmentation behavior of Armco iron and HF1 steel explosive filled cylinders, Journal of Applied Physics, 1983, Volume 54(5), 2344 – 2351.

  1. Line 98: Who does his refer to here?

We have changed the text as follows

“Mott concluded in his reference work [5]”

  1. Lines 159-162: This is not clear. Which assumption by Mott are the authors referring to? Please define time of metallic body fragmentation. Why is the characteristic term denoted by u_0^2?

Mott in his report, without providing explanations, assumes that the product between the square of the cylinder speed and a dimensionless ratio R whose value is given by the diameters of the explosive charge and the metal body is constant at the moment when the body is fragmented. In the analysis presented by Grady, he recognizes that the hypothesis proposed by Mott actually hides a principle of energy conservation. The demonstration presented in our work starts from energy conservation and we get at the end the same product introduced by Mott. In addition, in section 2.3 I analyzed the validity of this hypothesis using other indirect observations such as those related to the blast wave effect of ammunition. We also modified the text as follows and the term “time of metallic body fragmentation” was changed with “the moment when the metallic body is fragmented”

“Using the same assumption proposed by Mott [5] and recognized as a consequence of energy conservation by Grady [7], namely left-hand side of relation (8) has a constant value for the moment when the metallic body is fragmented”

The characteristic uo is introduced by Grady in his work. We had changed the text of the paper as follows

“the second-order radical of the right-hand side of the relation became a characteristic of the metal/explosive combination, denoted by uo

  1. Line 174: This study appears to be relying on a lot of assumptions from the cited work from Mott. Why is rt constant? Please provide an explanation and comment on if it is justified. And if it is indeed constant, why does that mean that the initial geometric data for the radius and wall thickness of the metal case will be used?

The product rt can also be expressed by the difference between the square of the outer diameter and the square of the inner diameter (see the paper text after the relation (10)). In the conditions in which it is assumed that the cylinder keeps its length and density during the expansion, this difference is constant as the cross section remains constant. We have changed the text as follows

“Since the density and the length of the metallic body are considered to be maintained constant during the radial expansion of the body, we get .. According to Mott, in this situation in formula (12) can be used the initial geometric data for the radius and wall thickness of the metal case [3].”

Regarding the use of the “the initial geometric data for the radius and wall thickness” we dedicated section 3.2 to this topic. Even if the exact value of the rt^5/6 product from the moment of fragmentation is not known, it can be approximated with the initial rt^5/6 product because the variation is small.

  1. Line 176-180: These are a lot of assumptions. Some of which don’t seem like they can be justified. Can the authors comment on these assumptions? It looks highly unlikely that the fracturing of one body does not influence the fracturing of the second body, especially if they are in such proximity?

        The hypothesis was analyzed in section 3.4 using numerical simulation as a study tool. A configuration with two cylinders was built in several variants, see Figure 5. The simulation for the normal variant shows that there is no complete lack of influence of the fragmentation processes, see the fractures that cross both cylinders. However, by combining the fragmentation results obtained in the additional studied variants, in which the fragmentation of one cylinder was blocked, and comparing them with the result obtained for the initial variant, it was shown that the mutual influence is not so important. Moreover, it was shown that the results of the simulation with this two cylinders can be easily approximated starting from the results of the simulation with a single cylinder of equivalent mass, by simply halving the scaling parameter from Mott's generalized law. This last result agrees with the observations from the section 4 for the configurations with cylinders of equal thickness.

The text of section 3.4 has been modified compared to the initial version, introducing references to the mutual influence, see the fractures that cross both cylinders, and to the method of determining the generalized Mott laws. Figure 6 was also modified and a new figure was introduced.

  1. Lines 192-194: Another assumption, can this be further justified?

        Section 3.1 is actually dedicated to the analysis of this hypothesis. For example, it is shown that for a metal mass/explosive mass ratio of 5, under the conditions of a cylindrical configuration, when the cylindrical body has doubled its average radius through expansion, the percentage difference between the speed of the inner and outer surfaces is only 8.7%. Under these conditions, if the body is actually made of two cylinders, the difference between the average radial speeds of these cylinders must be less than 8.7%, which is why we can approximate them as equal. It should not be forgotten that in the field of terminal ballistics, a field with a strong erratic character, the accepted differences are at least of the order of percentages.

  1. Line 424: The numbering is repeated at 5.

        The editing error has been corrected.

Reviewer 4 Report

1.     authors should add more reference in the introduction parts. In first 3 paragraphs, authors do not use any references.

2.     All the references are old (except of one that published in 2020), authors should update his references, use more recent publish paper.

3.     Novelty of the work should clearly mention in the introduction

4.     Why do researchers use the theoretical method with lot of assumptions, nowadays, we can achieve better result with numerical simulation. Please explain why we need theoretical method in introduction

5.     It was shown that the mott formula overestimates the number of large fragment, maybe using the modified mott formula help authors to achieve better result

“A Modification of the Mott Formula for Prediction of the Fragment Size Distribution”

6.     What is µ in equation 2 , please mention all of parameters and their units you use in all equations, readers should not go to find these parameters in paper that publish in 1943

7.      Please check the equation 1

Author Response

Dear Colleague,

first of all, I want to thank you for agreeing to analyze the paper written by our team. Also, thank you for the constructive comments related to the structure and its content, which we took into account in the revised form that was uploaded again on the platform. Next, I present to you how we treated each of your observations.

  1. authors should add more reference in the introduction parts. In first 3 paragraphs, authors do not use any references.

We had introduced two new citations

- Kritzinger, H. H.  Stuhlmann F. Artillerie und Ballistik in Stichworten, Berlin, Verlag von Julius Springer, 1939

- Mock, W.: Holt, W. H. Fragmentation behavior of Armco iron and HF1 steel explosive filled cylinders, Journal of Applied Physics, 1983, Volume 54(5), 2344 – 2351.

  1. All the references are old (except of one that published in 2020), authors should update his references, use more recent publish paper.

Some new references, from 2021-2022, relevant to the topic discussed, have been added to the list of references. One third of the list of references are published after 2010.

  1. Novelty of the work should clearly mention in the introduction

In the introduction section was added a new paragraph

“Even if the subject of natural fragmentation has been extensively analyzed by numerous researchers, there are no works dedicated to the derivation of specific formulas for multilayer structures like two or more coaxial metallic cylinders.”

  1. Why do researchers use the theoretical method with lot of assumptions, nowadays, we can achieve better result with numerical simulation. Please explain why we need theoretical method in introduction

In the introduction section was added a new paragraph

“In the last decades, with the increase in computing power, the numerical calculation methods have been imposed as an alternative to the traditional means of study. These methods can produce results faithful to reality if the material models are correctly chosen and the discretization of the bodies is appropriate [12-14]. However, the predetermined expeditious formulas, like Mott’s formula, do not require a consistent calculation effort nor to acquire special computer programming skills by the user, which is why they remain a tool that is still used as a benchmark in design calculations.”

  1. It was shown that the Mott formula overestimates the number of large fragment, maybe using the modified mott formula help authors to achieve better result

“A Modification of the Mott Formula for Prediction of the Fragment Size Distribution”

It does not yet require the use of modified formulas, considering the current stage of the subject addressed. When the experimental data obtained from the tests that we are going to carry out in the coming months will be available, this opportunity will also be analyzed in a future paper. Anyway, we consider that the recommended reference is one of interest. For this reason it was mentioned in the introduction section.

  1. What is µ in equation 2 , please mention all of parameters and their units you use in all equations, readers should not go to find these parameters in paper that publish in 1943

We added the definition of the parameter µ after equation (1) with the following text

“and µ the characteristic mass which represents half of the fragments average mass”

  1. Please check the equation 1

We checked equation 1. We did not use exactly the same notations as those from Mott's reports.

Round 2

Reviewer 1 Report

It is suggested to further verify the accuracy of the calculation model in this manuscript. Try to find the data by performing relevant tests or the data in recent literatures.

From Formula 18, Formula 19 and Figure 2, it is shown that the velocity of the inner surface vinis greater than the velocity of the outer surface vex. The explanation on this results cannot be proved unreasonably.

Please polish  furtherly.

Author Response

Dear Colleague,

thank you for the constructive comments related to the structure and content of our paper. Next, I present to you how we treated each of your observations.

1 It is suggested to further verify the accuracy of the calculation model in this manuscript. Try to find the data by performing relevant tests or the data in recent literatures.

As I wrote in the previous answer, the only reference with a similar approach is an article written by our team (Trană, E.; Bucur, F., Rotariu, A, N. On the fragmentation of explosively-driven plastic/steel layered cylinders, Materiale Plastice 2018, Volume 55(4), 521-523.)

The experimental data from that reference confirm the model presented there. In addition to those presented in the reference, the model used in this paper admits an independent fragmentation of the cylinders. This hypothesis benefits from an extended analysis in which numerical simulation is used as an analysis tool in section 3.4. A configuration with two cylinders was built in several variants, see Figure 5. The simulation for the normal variant shows that there is no complete lack of influence of the fragmentation processes, see the fractures that cross both cylinders. However, by combining the fragmentation results obtained in the additional studied variants, in which the fragmentation of one cylinder was blocked, and comparing them with the result obtained for the initial variant, it was shown that the mutual influence is not so important. Moreover, it was shown that the results of the simulation with this two cylinders can be easily approximated starting from the results of the simulation with a single cylinder of equivalent mass, by simply halving the scaling parameter from Mott's generalized law. This last result agrees with the observations from the section 4 for the configurations with cylinders of equal thickness. Also, the conclusion regarding the validity of this hypothesis is slightly nuanced.

As for conducting new experiments, even on a small scale, this activity requires a long working time of the order of months, but Materials journal requested us that the changes made to the content of the work to be carried out within 5 days of receiving your second review.

However, if you are aware of certain works that contain experimental results related to the subject we are addressing, we would be grateful if you would share this useful information with us to be included in our study and analysis.

  1. From Formula 18, Formula 19 and Figure 2, it is shown that the velocity of the inner surface is greater than the velocity of the outer surface vex. The explanation on this results cannot be proved unreasonably.

Our reasoning neglects the effects that the initial shock and rarefaction waves have on the displacement of the cylinder surfaces. To clarify this, we added the following text to the paper:

“when the density and the length of the metallic body are considered to be maintained constant and the effect of initial strong shock and rarefaction waves that act on the cylinder is neglected.”

We also decided to add an addition to the analysis performed in Section 3.1 based on the simulation results in Section 3.4. The results related to the speeds of the two surfaces, the inner and the outer, were analyzed. By introducing a new figure, the effects of shock waves and rarefaction in the initial phase of expansion have been highlighted, but also the fact that these effects disappear in time and at the moment of fragmentation the working hypotheses used in section 3.1 are valid. In the section 3.4 we added the following text:

“To complete the analysis given in section 3.1, in the simulation with two separate cylinders two gauges were defined. One was on the inner surface of the inner cylinder and the other one on the outer surface of the outer cylinder. The evolutions of the density and of the radial velocity on these two gauges were calculated and plotted in Figure 9. As can be seen, the first period, marked by the velocities strong oscillations and an important increase in density on the inner surface, corresponds to the occurrence and the propagation of shock and rarefaction waves back and forth through the cylinders walls. In the second period the density returns to values close to the initial value and the velocities keep an increasing trend with the speed of the inner surface being higher than the speed of the outer surface, a situation that corresponds to the relationships established in Section 3.1.

In the same figure the values of the average radial speeds of the cylinders are shown. Once the oscillations fade these two curves fall within the range determined by the speed of the inner surface and that of the outer surface. At the end of the plotted period the difference Vin - Vex is 8.7% of  velocity and the difference between the radial velocities of the cylinders is 3.3%. These results show that when the cylinders double their size, the value at which the cylinders are expected to fragment [17], the conditions used in the derivation of relations (18) and (19) are fulfilled.”

Reviewer 4 Report

Accept in present form

Author Response

Dear Colleague,

thank you for your peer-review.